# The Effects of Post-Warm-Up Active and Passive Rest Periods on a Vigilance Task in Karate Athletes

**DOI:** 10.3390/bs14111102

**Published:** 2024-11-15

**Authors:** Rui Miguel Silva, Francisco González-Fernández, Alba Rusillo-Magdaleno, Vânia Loureiro, Dinis Pires, Filipe Ferreira, Ana Filipa Silva

**Affiliations:** 1Escola Superior de Desporto e Lazer, Instituto Politécnico de Viana do Castelo, Rua Escola Industrial e Comercial de Nun’Álvares, 4900-347 Viana do Castelo, Portugal; rui.s@ipvc.pt (R.M.S.); filipe_ferreira1@hotmail.com (F.F.); abragasilva@esdl.ipvc.pt (A.F.S.); 2Research Center in Sports Performance, Recreation, Innovation and Technology (SPRINT), 4960-320 Melgaço, Portugal; 3Department of Physical Education and Sports, University of Granada, 18012 Granada, Spain; ftgonzalez@ugr.es; 4Department of Didactics of Musical, Plastic and Body Expression, University of Jaen, 23071 Jaén, Spain; arusillo@ujaen.es; 5Department of Arts, Humanities and Sports, School of Education, Polytechnic Institute of Beja, 7800-295 Beja, Portugal; 6Associação de Karate de Fafe, 4820-350 Fafe, Portugal; pires.a.dinis@gmail.com

**Keywords:** sports training, combat sports, youth, cognitive performance, reaction time

## Abstract

This study aimed to analyze how active versus passive rest periods after a warm-up influence performance in psychomotor vigilance tasks (PVT). Twenty amateur karate athletes participated in a randomized cross-over study consisting of two sessions with either a 20 min active rest involving kata techniques or passive rest. PVT was administered before and after these conditions to assess the changes in reaction time. The results revealed that the active rest condition significantly improved reaction times compared to both the passive rest condition (F(1,31) = 5.34, *p* = 0.03, η^2^ partial = 0.14) and control condition (F(1,31) = 5.49, *p* = 0.02, η^2^ partial = 0.15). No significant time-on-task effects were observed, F(4,120) = 2.31, *p* = 0.06, and there were no significant interactions between effort condition and time-on-task, F(4,120) = 1.89, *p* = 0.11). Participating in an active rest period post-warm-up improves cognitive performance in karate athletes, as evidenced by quicker reaction times in the PVT. This finding supports the use of active rest strategies (involving kata techniques) to maintain and improve cognitive readiness in young karate athletes.

## 1. Introduction

Karate is a martial art discipline that encompasses two primary forms of practice known as kumite and kata, demanding rapid reactions, simultaneous attack–defense actions, and refined motor skills [1]. Kata involves substantial mass displacements achieved by maintaining a lower center of gravity [2] and involves a predetermined sequence of movements performed with explosive speed, simulating engagements with imaginary opponents. On the other hand, kumite is characterized by high-intensity activity, including explosive and intermittent actions lasting from 0.3 to 2.1 s, demanding significant technical and tactical proficiency [3]. For those reasons, karate demands high levels of aerobic and anaerobic power capacity and strength, especially in the kumite discipline [4,5].

Although the physical fitness characteristics of karate athletes are determinant for such a combat sport, a great level of focus on the opponents’ actions is also an important requisite to ensure the athletes maintain a great level of sustained attention, especially in kumite training and competition, where there is opposition [6,7]. Indeed, in the kumite discipline, athletes have to sustain their attention on the opponent’s actions and selectively respond to relevant stimuli with speed and accuracy [8]. Sustained attention, often termed vigilance, is a cognitive function essential for maintaining attention through time [9]. Extended engagement in a task usually leads to mental fatigue and declines in a given task performance as time progresses [10]. As a consequence, the reduced levels of sustained attention lead to delayed reaction times, decreased anticipation, and a greater inability to detect a given target [11,12,13]. For instance, previous studies showed that attention typically diminishes within the timeframe of 10 to 30 min into a task [14,15].

Despite the absence of evidence regarding the duration between the warm-up and the start of official combat in karate kumite competitions, karatekas often warm up without knowing when their first combat will take place [16]. Therefore, karate athletes are obligated to be in a passive rest condition from the warm-up until the start of the first combat, which can take several minutes to one hour of waiting [7,16]. Extended periods between the warm-up and the competition are expected to result in a greater dissipation of the warm-up effects [17]. For instance, a systematic review of the time between warm-up and match start in team sports found that a 2 min active re-warm-up with short explosive movements is necessary for intervals longer than 15 min [17]. The warm-up serves as post-activation performance enhancement (PAPE), which refers to the acute stimulation of the neuromuscular system after engaging in a given exercise to improve subsequent performance [18,19]. PAPE, often achieved through high-intensity exercises such as plyometric exercises, is designed to enhance power output and explosiveness in subsequent activities [20]. However, it is important to note that many contemporary warm-up routines incorporate dynamic activities, such as jumping or sprinting drills, not solely to achieve PAPE but also to optimize the warm-up effect through specificity and skill rehearsal, ensuring athletes are well-prepared for sport-specific movements [21,22]. This phenomenon of PAPE, however, typically lasts only for several minutes [23].

Furthermore, it has been shown that physical exercise improves cognitive performance [24]. A study that analyzed the role of aerobic fitness in sustained attention measured by a psychomotor vigilance task (PVT), which measures the reaction time to respond to a given stimulus, showed faster responses for the group with greater aerobic capacity [25]. Indeed, a previous systematic review that analyzed the effects of physical activity on sustained attention showed that participating in exercises from moderate to high-intensity results in improved sustained attention [26]. Indeed, active breaks, consisting of strength and self-loading exercises, have been shown to improve performance in PVT tasks [27]. Moreover, a recent study conducted on thirty-two students to analyze the effects of the inclusion of a warm-up on a PVT task showed that participating in a warm-up improved sustained attention as compared to the group that received no warm-up [28]. Also, research has shown that adolescent athletes are particularly responsive to cognitive and physical interventions, with their neuromuscular and attentional capacities developing in ways that are still distinct from adults [29]. For young athletes engaged in combat sports, improving sustained attention and physical readiness through structured warm-ups and rest protocols could provide meaningful benefits not only in performance but also in developing long-term mental and physical resilience [30].

Given that there is no evidence of the effects of post-warm-up active and passive rest periods on PVT performance in karate and that karate athletes usually wait for longer periods between the warm-up and the first combat, the present study aims to analyze the effects of an active and passive rest period after a warm-up on the PVT and to compare the differences between the active and passive rest periods on the PVT. We hypothesize that an active rest period following the warm-up will lead to significantly better performance on the PVT, as compared to passive rest, by helping to sustain the neuromuscular and cognitive benefits associated with the active rest activities (e.g., kata techniques).

## 2. Materials and Methods

### 2.1. The Study Design and Experimental Approach

This present study employed a randomized cross-over design. Participants were recruited from two different karate teams. The psychomotor vigilance task (PVT) was conducted over two sessions, with a one-week interval between them. Participants were randomly assigned to either the passive rest group, active rest group, or control group using a randomization sequence that was generated electronically and was concealed until the interventions were assigned. The passive rest group completed a five-minute PVT both before and after the warm-up. Then, after a 20 min interval after the second PVT, the group completed a third five-minute PVT. The active rest group completed a five-minute PVT both before and after the warm-up. Then, they immediately engaged in a 20 min physical activity consisting of karate kata techniques. After the 20 min karate kata techniques, the active rest group completed a third five-minute PVT. The control group completed only the five-minute PVT without participating in any form of physical activity (i.e., the warm-up and the kata techniques). The following training session, held a week later, saw the roles reversed.

Before the main experimental tasks, all participants underwent a familiarization session to ensure they understood the procedures and to minimize any learning effects that could influence the results. This session included a brief orientation on the PVT protocol, where participants completed a practice five-minute PVT trial under the same conditions as the actual testing to acquaint themselves with the response timing and concentration requirements of the task. Familiarization with the kata techniques was also provided to those in the active recovery group, with supervised practice to ensure proper form and intensity. No trials were excluded from the dataset.

### 2.2. Participants

Twenty karate amateur athletes (age: 15.7 ± 3.3 years; height: 169.2 ± 7.3 cm; body mass: 65.1 ± 5.7 kg) participated in this study. The inclusion criteria for participant selection were as follows: (i) previously signing the provided informed consent form by their legal guardians; (ii) being in good health without any medical contraindications or musculoskeletal injuries, regardless of gender; (iii) attending the two training sessions; and (iv) participating in the two groups. All participants were provided with detailed information regarding the study design, potential risks, and benefits and provided voluntary written informed consent to participate before the study commencement. This study was conducted following the Ethics Committee for Social, Life, and Health Sciences of the Polytechnic Institute of Viana do Castelo with the code CECSVS2024/02/I and in compliance with the principles outlined in the Declaration of Helsinki.

### 2.3. Psychomotor Vigilance Task (PVT)

An iPhone 5s operating on iOS version 12.4.5 was used to deliver the stimuli for the Psychomotor Vigilance Test (PVT) app (Vigilance Buddy app, version 1.57). Before the experimental sessions, these devices were configured to disable all notifications to prevent any distractions that could interfere with the participants’ focus. The mobile device was strategically positioned at eye level, approximately 50 to 80 cm from the participants’ heads, ensuring optimal visibility and engagement with the screen. The PVT interface featured a gray background with a chronometer prominently displayed at the center.

The test commenced after a randomly determined interval, which varied between 2000 and 10,000 ms. During this interval, the chronometer functioned similarly to a real stopwatch, providing a visual cue for the participants. Before initiating the PVT in each session, participants received both verbal and written instructions emphasizing the importance of maintaining fixation on the center of the screen. They were instructed to minimize eye movement and to respond as swiftly as possible once the chronometer began to fill while also being cautioned against making anticipation errors. Participants recorded their responses during a single block lasting five minutes by pressing the center of the device as quickly as possible when the stopwatch started. Following the protocol previously established [31], trials with reaction times below 100 milliseconds were classified as anticipation errors and subsequently excluded from the analysis. The exact number of attempts per participant depended on the individual’s response latency. The outcomes extracted from the data were the mean reaction times, in milliseconds (ms), during the five-minute PVT of each participant trial across conditions (passive rest, active rest, and control groups), providing a comprehensive assessment of their performance.

This meticulous approach not only enhances the reliability of the findings but also contributes to a deeper understanding of the impact of warm-up protocols on the PVT. By analyzing the mean reaction times, researchers can draw meaningful conclusions regarding the cognitive and motor performances of participants under varying conditions, thereby informing future studies and practical applications in sports science and psychology. The reaction times were recorded for each PVT in CSV format on the respective devices and subsequently emailed to the experimenter.

### 2.4. Statistical Analysis

A two-way repeated measures ANOVA was conducted to assess the main effects of the effort condition (control, active rest, and passive rest) and time-on-task, defined as the sequence of five-minute psychomotor vigilance task (PVT) trials administered at different stages (i.e., before the warm-up, immediately after the warm-up, and after the assigned 20 min passive and/or active rest condition), as well as the interaction between these factors on the mean reaction times. Reaction time, the primary outcome variable, was measured in milliseconds (ms) and represents the average time taken by the participants to respond to visual stimuli during each five-minute PVT session. Partial eta-squared values (η^2^ partial) were reported to estimate the effect sizes for each effect, with the interpretation based on Cohen’s conventions [32]: small (0.2), medium (0.5), and large (>0.8). Post hoc comparisons were conducted with Bonferroni adjustments to identify specific group differences in the reaction times when significant effects were detected. Assumptions underlying the ANOVA were tested to ensure the validity of the analysis. The normality of reaction times was evaluated using the Shapiro–Wilk test, while the homogeneity of variances across conditions was assessed with Levene’s test. Mauchly’s test was used to evaluate sphericity for within-subject factors, and when this assumption was violated, the Greenhouse–Geisser correction was applied, with the adjusted *p*-values reported. Additionally, reaction times below 100 milliseconds were considered anticipation errors and excluded from the analysis to prevent artificially inflated performance scores, with the total number of valid trials reduced accordingly. Measures of central tendency (mean) and dispersion (standard deviation) were calculated and presented for each effort condition. To further enhance interpretability, 95% confidence intervals (CI) were calculated for each mean reaction time. All statistical analyses were performed using SPSS software (version 27.0), with the statistical significance set at *p* < 0.05.

## 3. Results

Repeated measures analyses of variance (ANOVA) were conducted using the average reaction times of the participants across the different conditions (Figure 1).

The results for each condition were as follows: control condition (395 ms ± 32 ms; 95% CI = 39.4, lower CI = 355, upper CI = 434), active rest condition (360 ms ± 108 ms; 95% CI = 22.0, lower CI = 338, upper CI = 382), and passive rest condition (381 ms ± 126 ms; 95% CI = 28.7, lower CI = 353, upper CI = 410). The time-on-task for each trial was set at 5 min. These analyses aimed to elucidate the effects of different conditions on participants’ reaction times, providing insights into the impact of warm-up activities on cognitive performance.

Repeated measures analyses of variance (ANOVA) were performed using the participants’ average reaction times across the conditions: control condition (395 ms ± 32 ms, 95% CI = 39.4, lower CI = 355, upper CI = 434), active rest condition (360 ms ± 108 ms, 95% CI = 22.0, lower CI = 338, upper CI = 382), and passive rest condition (381 ms ± 126 ms, 95% CI = 28.7, lower CI = 353, upper CI = 410); and time-on-task (5 min).

An ANOVA was conducted to compare participants’ mean reaction times in the control condition and active rest condition, as well as time-on-task. The analysis revealed a significant main effect of effort condition, F(1,31) = 5.5, *p* = 0.02, η^2^ partial = 0.15. Participants exhibited faster response times in the active rest condition (360 ms ± 108 ms) compared to the control condition (395 ms ± 32 ms). However, there was no significant main effect of time-on-task, F(4,120) = 2.1, *p* = 0.08, η^2^ partial = 0.07, nor a significant interaction between the effort condition and time-on-task, F(4,120) = 1.9, *p* = 0.11, η^2^ partial = 0.06.

In a separate ANOVA comparing the mean reaction times between the control condition and passive rest condition, no significant main effects were found for either the effort condition, F < 1, or the interaction between the effort condition and time-on-task, F < 1. However, the analysis revealed a significant main effect of time-on-task, F(4,120) = 4.7, *p* = 0.001, η^2^ partial = 0.13, indicating a decrease in the reaction times over time.

Finally, an ANOVA comparing the mean reaction times between the active rest condition and passive rest condition revealed a significant main effect of the effort condition, F(1,31) = 5.3, *p* = 0.03, η^2^ partial = 0.14. Participants demonstrated quicker responses in the active rest condition (360 ms ± 108 ms) compared to the passive rest condition (381 ms ± 126 ms). However, there was no significant main effect of time-on-task, F(4,120) = 2.3, *p* = 0.06, η^2^ partial = 0.06 (Table 1).

## 4. Discussion

This present study aimed to investigate the effects of post-warm-up active versus passive rest periods on the psychomotor vigilance task performance of karate athletes. The findings showed that an active rest period involving kata techniques significantly enhanced the athletes’ cognitive performance, as evidenced by the quicker reaction times compared to both the passive rest and control conditions. Additionally, no significant time-on-task effects or interactions between the effort condition and time-on-task were observed, suggesting that the improvements in reaction times were specifically due to the active rest period rather than fatigue or task duration.

It was observed that brief periods of moderate physical activity, such as active rest involving karate kata techniques, led to significant improvements in cognitive performance, particularly in a task requiring focused attention. The results of the present study corroborate previous research, indicating that brief periods of moderate physical activity can lead to immediate improvements in cognitive tasks, especially those requiring focused attention and executive function [33]. Adding to this, recent evidence suggests that adults with martial arts experience exhibit enhanced performance in maintaining endogenous alertness, a cognitive domain closely linked to attention [34,35]. However, the mechanisms by which this occurs in martial arts practitioners might be multifaceted and warrant further exploration. In the context of martial arts, particularly in karate kata techniques, these improvements might not only be due to increased physiological arousal but could also involve psychological components such as the meditative state often associated with such techniques [35]. This aligns with the observed benefits in attentional networks, where martial arts training could improve the ability to sustain attention [34]. This physiological and psychological duality could facilitate greater neural connectivity, particularly in the prefrontal cortex [36]. Additionally, the developmental stage of the participants may have played a role in enhancing neuroplasticity, which supports cognitive maturation in adolescents and young adults [37,38]. Future research could investigate whether these cognitive improvements are specifically linked to certain martial arts actions, such as the precision of movements, and could benefit from using neuroimaging techniques to explore the changes in brain activity patterns.

The lack of significant time-on-task effects suggests that the cognitive benefits of active rest are maintained throughout the task, which could be particularly beneficial in sports such as karate, where sustained attention over time is crucial [39]. This finding is supported by research conducted on high school students, where an eight-week program of active breaks resulted in improved vigilance performance, indicating that regular, short bursts of physical activity can improve cognitive function over time [27]. The same authors [27] suggested that the integration of physical activity during breaks could serve as an effective strategy for sustaining cognitive performance not only in educational settings but also in other sports. However, in our study, we considered the acute effects of active rest and passive rest conditions on a vigilance task, while in the above-mentioned study, ref. [27] implemented a chronic response design. Regarding the acute effects, a previous study found that vigilance performance improved after exercise at 80% of the ventilatory anaerobic threshold during short-term tasks [40]. Additionally, sustained performance benefits were observed when exercising at 75% of the ventilatory anaerobic threshold over longer durations [40]. This is the influence of exercise intensity on reaction times during vigilance tasks. This is particularly valuable in martial arts competitions, where maintaining good levels of cognitive function over extended periods is essential, as athletes often experience delays of up to one hour between their warm-up and the start of their first combat [39]. The sustained effectiveness could also be attributed to the combination of physical movement with mental focus in kata, which might help in maintaining neural efficiency without leading to cognitive fatigue [41]. Future studies could explore the optimal duration and frequency of active rest periods to maximize sustained attention.

The finding that passive rest did not significantly improve vigilance performance compared to the active rest condition supports the hypothesis that sustained attention and better reaction times require an active physical and/or cognitive task. This aligns with findings from previous studies where active breaks involving moderate physical exercises during classes in high school students and schoolchildren of 10–11 years led to improved vigilance performance [27,42,43]. Moreover, the kata techniques, with their choreographed movements, might stimulate not only physical outputs but also require a great level of mental visualization [44]. Furthermore, research on martial arts training indicates that prolonged engagement in activities demanding sustained attention can specifically improve the alertness network in adults, illustrating how activities such as kata could contribute to cognitive benefits through sustained attention mechanisms [35]. This difference between passive rest and active rest could be critical in designing interventions for vigilance and sustained attention, where activities combining physical movement with cognitive tasks might be more effective than passive rest.

This present study has its limitations. The intensity of the active rest was not monitored, which could have influenced the observed outcomes. Future research could benefit from quantifying both the subjective and objective measures of intensity to better understand how varying levels of intensity impact cognitive performance. Moreover, the baseline physical fitness and mental condition of participants might influence the observable benefits of short-term interventions such as active rest in our study. Given the lack of significant findings in this area, future studies could explore how participants with differing levels of physical fitness and mental readiness respond to active and passive rest periods. Comparative studies involving participants from a broader range of fitness and cognitive statuses could provide deeper insights into how baseline fitness and mental status influence the cognitive effects of different resting periods.

## 5. Conclusions

In conclusion, the findings of this study showed that implementing kata techniques during active rest after a period of moderate physical exertion, such as a warm-up, improves the cognitive performance of karate athletes, demonstrated by faster reaction times relative to the conditions of passive rest and the control groups. Given that, active rest strategies after a period of physical exertion help to maintain and improve the reaction times during a vigilance task. These findings are particularly relevant for karate competitions, where athletes often experience delays of up to one hour between their warm-up and the start of their first combat. From a neurophysiological perspective, active rest may help sustain neural activation and cognitive performance by maintaining neuromuscular engagement and facilitating recovery.

## Figures and Tables

**Figure 1 behavsci-14-01102-f001:**
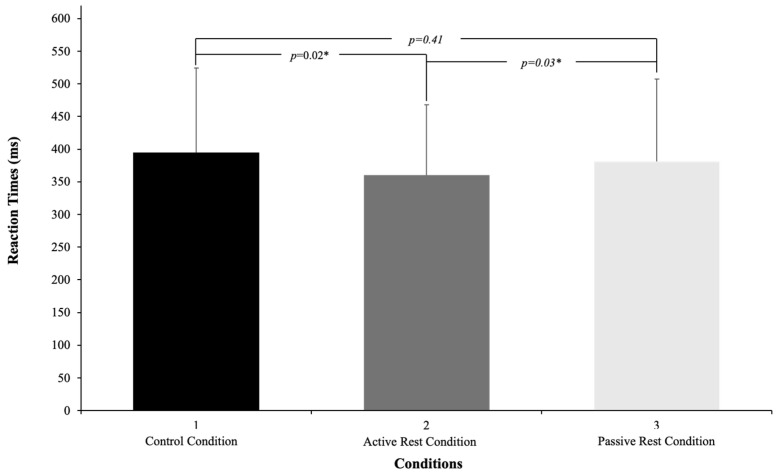
PVT participants’ mean RT (±SE) and individual performance as a function of condition. * denotes significance at *p* < 0.05.

**Table 1 behavsci-14-01102-t001:** Repeated measures analyses of variance (ANOVA) for the main effects of effort condition and time-on-task.

Comparison	Condition	Reaction Time (Mean ± SD)	95% CI [Lower; Upper]	*F*-Value	*p*-Value	*η*^2^ Partial
Control vs. Active Rest	Control	395 ± 32	39.4 [355; 434]	5.5	0.02	0.15
	Active Rest	360 ± 108	22.0 [338; 382]			
Control vs. Passive Rest	Control	395 ± 32	39.4 [355; 434]	<1	0.01	0.13
	Passive Rest	381 ± 126	28.7 [353; 410]			
Active Rest vs. Passive Rest	Active Rest	360 ± 108	22.0 [338; 382]	5.3	0.03	0.14
	Passive Rest	381 ± 126	28.7 [353; 410]			

## Data Availability

The original contributions presented in the study are included in the article, further inquiries can be directed to the corresponding author.

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
