# Peer review of "The Effects of Post-Warm-Up Active and Passive Rest Periods on a Vigilance Task in Karate Athletes"

_behavsci, 2024, doi:10.3390/bs14111102_

Round 1
Reviewer 1 Report
Comments and Suggestions for Authors
The study is original and interesting. Please consider my specific comments below for quality improvement of your manuscript.
Specific comments
Line 30: Please provide some examples (in brackets) of the active recovery methods used.
Lines 58-59: Please add some support or references here.
Line 64: The point is interesting, but you are now discussing “re-warm up,” which is usually used during e.g., half-time in football matches. You need to provide a clearer connection here.
Line 64: "A previous systematic review" – Please delete "previous."
Lines 66-68: It is important to expand more on PAPE here. Note: Many warm-up sessions nowadays incorporate jumping activities, not necessarily to benefit optimally from PAPE but more for optimising "warm-up effect" through e.g., specificity and skill rehearsal.
Line 85: It seems a hypothesis is missing. Please consider adding one.
Line 88: Remove "The."
Line 89: Can you provide more detail about the familiarization with the tasks involved?
Line 105: The subject is young/adolescent. I believe this cohort needs to be discussed or elaborated on in the introduction.
Line 117: Is there any information about the camera or frame rate used, along with the setup details?
Line 150: Which type of ANOVA was employed? Please provide more details.
Line 153: Is there any information regarding the interpretation or magnitude of the effect sizes used?
Lines 156-158: This sentence is unclear. Can you clarify?
Lines 162-165: For readability, please remove all decimals (e.g., 394.69 ms should be written as 395 ms). Retain one decimal only if the value is less than 10. Please check for consistency throughout the paper.
Line 196: The standard deviation seems large. Is this discussed in the discussion section? Or are the bars not presented appropriately? Did you start at 320?
Line 201: "Karate athletes" should be plural, correct?
Line 202: Replace the word "potentiate" with "increase."
Line 203: What else did you find? And please conclude the sentence clearly.
Line 205: I disagree with the current structure. Please begin the paragraph with the most important result, followed by critical discussion.
Lines 217-221: These two sentences are unclear and seem disconnected. Can you revise for clarity?
Line 222: This paragraph lacks any discussion of the role of warm-ups in cognitive performance.
Lines 234-236: The assertion here is unclear. Can you rephrase it for clarity?
Line 246: Which hypothesis does this support? Please specify (do you even have a hypothesis?)
Lines 250-252: Please support this statement. Are you suggesting that passive rest prevents the use of mental visualization?
Line 265: Since the results lack significance, please suggest future research directions to explore this topic further.
Final Comment: I recommend avoiding abbreviations like "RT." Please write terms out in full.
Comments on the Quality of English Language
As noted above
Reviewer 2 Report
Comments and Suggestions for Authors
Well-structured research. An interesting topic was investigated for several sports, which has an impact on sports performance.
The study provides relevant information for the preparation of athletes in competition (karate), which may help to improve their results
The introduction adequately addresses the empirical background and conceptualization of the study variables.
Well-developed method and procedure, they clearly explain how the data was obtained in the different measurements.
Discussion, properly developed, allow for an in-depth analysis of the subject. To visualize the achievements and potential of the results.
The conclusions are appropriate, respond to the objective of the study and highlight the importance of its practical application for karate competitions.
It is also required to be modified:
Statistical analysis section, it should be written specifically in relation to the analysis carried out. This is to facilitate the reader's reading and understanding. The current wording complicates understanding.
In results, including one graph, it is insufficient to give an account of the results obtained. It is necessary to include one or two tables, to better visualize the values obtained in each measurement and variable of the study, currently reading a sheet without a table makes it difficult to clearly understand this point.
Reviewer 3 Report
Comments and Suggestions for Authors
Dear authors,
Your article proposal on the influence of active and passive rest periods after a warm-up on performance during a psychomotor vigilance task (PVT) is relevant and addresses major concerns for optimizing efficiency in competition, even beyond the discipline of karate.
To strengthen the scientific rigor of the article, it would be useful to describe the experimental protocol more precisely. We note that the study focuses on a population of 20 young amateur karate athletes. However, it would be preferable to detail the structuring of the warm-up and the procedure for test success. For example, if the warm-up is collective, the benefits may differ between the first and the last athlete to take the test, which could influence the results. Adding a descriptive figure of the protocol would also be beneficial to clarify the functionality and organization of individual execution.
Furthermore, you mention key stages of reaction time, such as perception, information processing, and motor response. Does your protocol allow for distinguishing these different phases? These aspects are often influenced by factors such as fatigue, mood, or attention, which modulate reaction speed. A clarification on the nature of cognitive tasks and their link to these stages would be appreciated.
On line 138 (p.3), a distinction between cognitive and motor performances could be further clarified. Finally, although you address this point, could you enrich the conclusion by delving into the neurophysiological perspectives related to the effects of warm-up and rest periods (active and passive) on psychomotor performance?
Best wishes.
Round 2
Reviewer 1 Report
Comments and Suggestions for Authors
The authors have revised their manuscript accordingly. Thank you
Comments on the Quality of English LanguageI think that the manuscript will benefit from professional English edits.
Author Response
The authors have revised their manuscript accordingly. Thank you.
I think that the manuscript will benefit from professional English edits.
AUTHORS: Dear Reviewer, thank you for your comments. In the first round of revisions, we sent our manuscript to a native English colleague who edited the entire manuscript language. Also, in the first round cover letter to you, we wrote the following at the end of the letter: "We made improvements to the English language throughout the entire manuscript to enhance grammatical accuracy, readability, and clarity."